# Modelling the incremental benefit of introducing malaria screening strategies to antenatal care in Africa

Patrick G. T. Walker [1✉], Matt Cairns [2], Hannah Slater[1,3], Julie Gutman [4], Kassoum Kayentao[5], John E. Williams[6], Sheick O. Coulibaly[7], Carole Khairallah[8], Steve Taylor[9], Steven R. Meshnick[10], Jenny Hill [7], Victor Mwapasa [11], Linda Kalilani-Phiri[9], Kalifa Bojang[12], Simon Kariuki [13], Harry Tagbor [14], Jamie T. Griffin[15], Mwayi Madanitsa[11], Azra C. H. Ghani[1], Meghna Desai[4] & Feiko O. ter Kuile[7]

*Plasmodium falciparum* in pregnancy is a major cause of adverse pregnancy outcomes. We combine performance estimates of standard rapid diagnostic tests (RDT) from trials of intermittent screening and treatment in pregnancy (ISTp) with modelling to assess whether screening at antenatal visits improves upon current intermittent preventative therapy with sulphadoxine-pyrimethamine (IPTp-SP). We estimate that RDTs in primigravidae at first antenatal visit are substantially more sensitive than in non-pregnant adults (OR = 17.2, 95% Cr.l. 13.8-21.6), and that sensitivity declines in subsequent visits and with gravidity, likely driven by declining susceptibility to placental infection. Monthly ISTp with standard RDTs, even with highly effective drugs, is not superior to monthly IPTp-SP. However, a hybrid strategy, recently adopted in Tanzania, combining testing and treatment at first visit with IPTp-SP may offer benefit, especially in areas with high-grade SP resistance. Screening and treatment in the first trimester, when IPTp-SP is contraindicated, could substantially improve pregnancy outcomes.

[1] MRC Centre for Global Infectious Disease Analysis, Department of Infectious Disease Epidemiology, Imperial College London, London, UK. [2] London School of Hygiene and Tropical Medicine, London, UK. [3] PATH, Seattle, WA, USA. [4] Malaria Branch, Division of Parasitic Diseases and Malaria, Center for Global Health, Centers for Disease Control and Prevention, Atlanta, GA, USA. [5] Malaria Research and Training Centre, Department of Epidemiology of Parasitic Diseases, Faculty of Medicine, Pharmacy, and Dentistry, University of Sciences, Techniques, and Technologies of Bamako, Bamako, Mali. [6] Dodowa Health Research Centre, Dodowa, Ghana. [7] Faculty of Health Sciences, University of Ouagadougou, Ouagadougou, Burkina Faso. [8] Department of Clinical Sciences, Liverpool School of Tropical Medicine, Liverpool, UK. [9] Global Health Institute, Duke University, Durham, NC, USA. [10] University of North Carolina, Chapel Hill, NC, USA. [11] College of Medicine, University of Malawi, Blantyre, Malawi. [12] Medical Research Council, London School of Hygiene and Tropical Medicine, Fajara, The Gambia. [13] Kenya Medical Research Institute/Centre for Global Health Research, Kisumu, Kenya. [14] University of Health and Allied Sciences, Ho, Ghana. [15] School of Mathematical Sciences, Queen Mary University of London, Mile End Road, London, UK. ✉email: patrick.walker@imperial.ac.uk

nfection with *Plasmodium falciparum* malaria in pregnancy (MiP) is associated with a wide range of adverse pregnancy outcomes including maternal anaemia, low birthweight and neonatal death[1]. These adverse effects largely result from sequestration of the parasite within the placenta particularly in women not exposed to *P. falciparum* in any previous pregnancy[1]. Despite declines in malaria transmission in many settings[2], MiP risk remains high[3]. Approximately a third of all pregnancies (9.5m of 30.6m) occurring in areas of sustained transmission in 2015 were liable to be affected by malaria[3]. In the absence of pregnancy-specific protection, this could lead to 750,000 malaria-attributable low birthweight deliveries in sub-Saharan Africa each year[3]. Observed increases in the average density of placental infection in areas where transmission has fallen suggest declining immunity will ensure MiP continues to represent a pressing public health concern even if the current stall in reducing global malaria transmission is overcome[4–6].

Despite significant improvements in access to antenatal care (ANC) in the past decade, uptake of proven effective tools for MiP prevention has been slow. In areas where intermittent preventative therapy in pregnancy (IPTp) is recommended, 22% of women received the recommended three or more doses of IPTp in 2017[6]. Moreover LLINs use is low in adolescents who are the most at risk of high-density placental infection[3]. The emergence of parasite resistance to sulfadoxine–pyrimethamine (SP), the only drug currently recommended by the World Health Organisation (WHO) for IPTp, has led to attempts to find alternative strategies. One such alternative, intermittent screening and treatment in pregnancy (ISTp), has been evaluated in a number of countries[7–10]. Whilst, IPTp provides SP to all women at each visit without testing, ISTp involves testing of all pregnant women regardless of the presence of malaria symptoms (screening) with rapid diagnostic tests (RDTs) and treating test-positive women with highly efficacious artemisinin combination therapy (ACT). However, ISTp using the current generation of RDTs has not proven more effective than IPTp-SP[11], and is therefore not recommended by WHO[12].

WHO has recommended further studies into alternative strategies involving routine screening for MiP[12]. These include evaluating whether more sensitive RDTs could make ISTp a viable alternative and whether hybrid strategies, involving adding RDT-based screening to existing IPTp regimens, provide additional benefit[13]. One such hybrid approach is now national policy in Tanzania (where quintuple mutants are ubiquitous and highly resistant sextuple mutants have been identified in some areas): all women are tested for malaria parasites at the first ANC visit (booking) and provided with an ACT if test-positive or, starting in the second trimester, with SP if test-negative. All women then receive IPTp-SP at subsequent scheduled ANC visits[14]. However, given the additional costs and complexities of such approaches, it is important to understand where, and the extent to which, they can provide greater protection from malaria during pregnancy than standard IPTp-SP regimens.

The negative consequences of first trimester infection[11,15], at which time the use of SP is contra-indicated, and reassuring data regarding first trimester safety of ACTs[16] has resulted in increased interest in screen and treat approaches using ACTs for women attending ANC during the first trimester. A recent study from Benin tracking women prior to conception supports previous modelling[17], which suggested that a high proportion of placental infections are likely to be caused by infections acquired prior to conception. Genotyping of infection showed that the densities of infection acquired prior to pregnancy, rather than declining over time, had substantially increased by the time the women attended ANC[18].

Trials assessing the impact of MiP interventions are expensive, and time-consuming, requiring up to 2 years longitudinal follow-up. Moreover, large sample sizes are required to adequately measure effectiveness and cost-effectiveness as the key drivers of attributable burden are outcomes, such as pregnancy loss, low birthweight and neonatal mortality are increasingly rare in the context of clinical trials. Although not a substitute for clinical trials, modelling provides a means to explore the potential of multiple alternative interventions to guide prioritisation of research.

A key determinant of the need for alternatives to IPTp-SP, the level of parasite resistance to SP and the associated decline in efficacy of IPT-SP[19], varies across Africa. In much of West Africa, SP provides near perfect curative efficacy and a period of prophylaxis of approximately one month. In East Africa, where there is a very high prevalence of parasites harbouring K540E 'quintuple' SP resistance mutation, SP fails to clear ~20% of infections during pregnancy and provides limited prophylaxis[20]. Though there have been no efficacy studies in areas of high prevalence of the 'sextuple' SP resistance mutation (an additional mutation at A581G on top of the quintuple), currently limited to specific foci in East Africa[21], there are concerns that IPTp-SP effectiveness may be heavily compromised within these settings[19].

In this analysis, we combine data on the sensitivity of standard RDTs during pregnancy collected during ISTp trials with equivalent data from a review of RDT sensitivity outside of pregnancy[22]. We then use modelling to estimate the impact of pregnancy on the detectability of infection using RDTs, incorporating the role of pregnancy-specific immunity in controlling parasite densities in the placental and peripheral blood. Finally, we incorporate these estimates within a model of the relationship between malaria transmission and effectiveness of IPTp-SP, incorporating the effects of SP resistance, to assess the potential for different strategies involving antenatal screening, either with current or more sensitive RDTs, to improve protection for pregnant women from MiP.

## Results

**Impact of pregnancy upon detectability of infection by RDT.** There have been four large-scale trials comparing ISTp with IPTp-SP, three of them had matched RDT (First Response Malaria pLDH/HRP2 Combo Test, Premier Medical Corporation, India) and PCR samples collected from 1559 women based in six countries (Burkina Faso, The Gambia, Ghana, and Mali in West Africa and Kenya and Malawi in East Africa)[8–10]. West African studies recruited only women in their first and second pregnancy, whereas the studies in East Africa recruited women of all gravidities. In all studies women were enroled in their first visit after 16 weeks gestation provided this visit was before 28 weeks, 30 weeks and 32 weeks in Malawi, Kenya and West Africa, respectively. In three countries, infection by conventional RDT and PCR was measured throughout pregnancy: Ghana, Kenya and Malawi (in Burkina Faso, The Gambia and Malawi PCR was only measured at enrolment). Both prevalence and detectability (measured by RDT sensitivity relative to PCR) were consistently higher at enrolment than at subsequent ANC visits, particularly in primigravidae (Fig. 1).

In all six countries, RDT sensitivity at enrolment, defined throughout this paper as the level of detection relative to PCR, showed a declining trend with gravidity (Fig. 2a). Overall sensitivity in primigravidae was very high, with 88.9% [640/720, 86.4–91.1% 95% CI] of PCR-positive infections detected by RDT, but showed substantial heterogeneity between sites ranging between 65.8% [27/41, 49.4–79.9% 95% CI] sensitivity in Bassé, The Gambia, the setting with lowest transmission (PCR

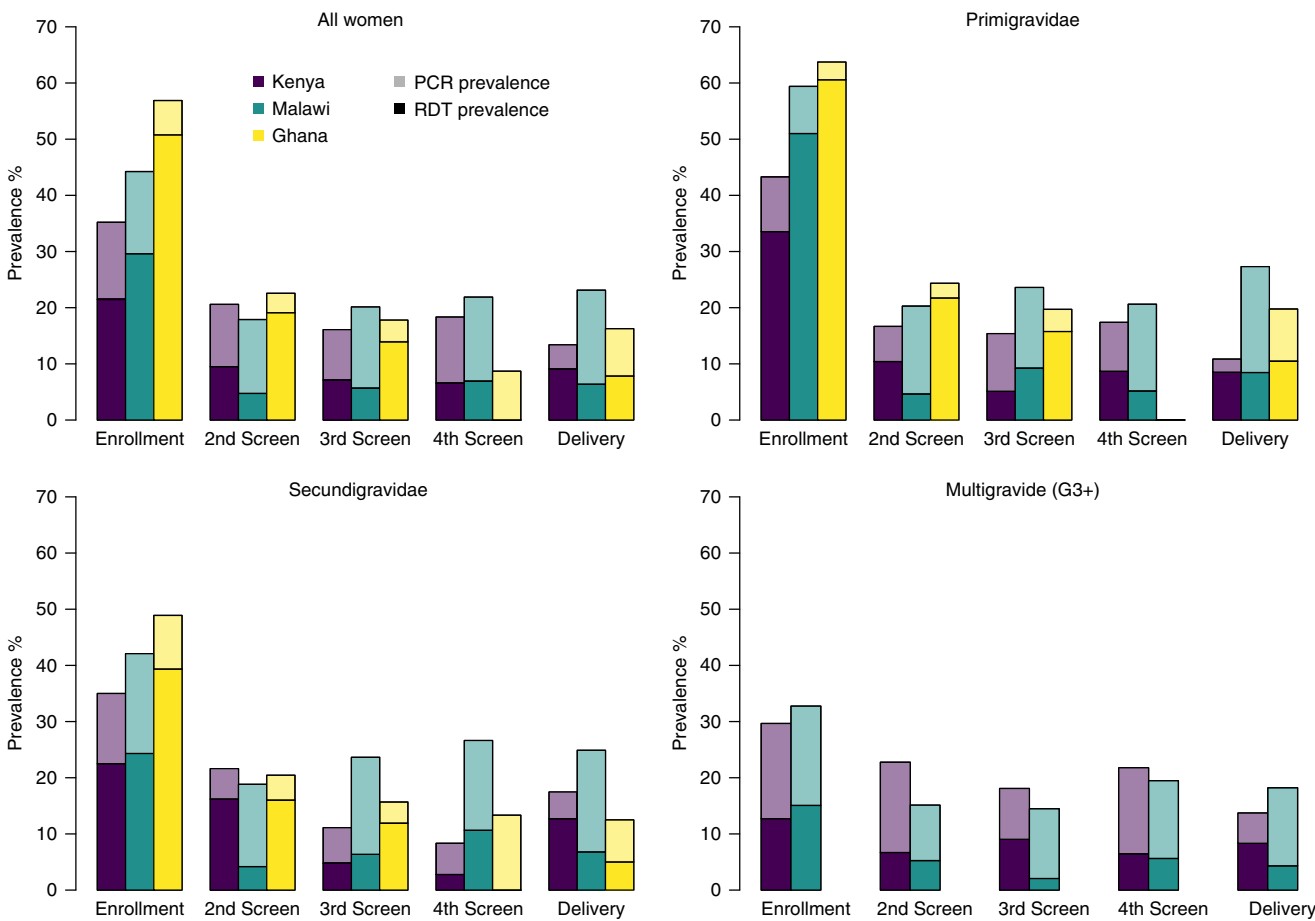

**Fig. 1 Prevalence by and RDT and PCR during ISTp.** Figure shows prevalence of RDT-positive infection, confirmed by PCR (height of darker bars) and the additional prevalence of RDT negative, PCR positive infection prevalence (height of lighter bars) at each ANC visit at which RDT testing was carried out during ISTp from enrolment to delivery by trial site.

prevalence in primigravidae: 13.4% [41/316, 9.8–17.8% 95% CI]), to close to that of PCR in Navrongo, Ghana (95.0% [192/202, 91.1–97.6% 95% CI]), where PCR prevalence was 65.8% [202/307, 60.2–71.1% 95% CI]. We also compared the sensitivity observed within primigravidae at enrolment against the RDT sensitivity in other non-pregnant populations from data obtained from a recent review[22]. Incorporating the relationship between transmission and RDT sensitivity from this analysis (Fig. 2), we estimated that the odds of detecting a PCR-positive infection with an RDT were substantially higher at enrolment in primigravidae than in asymptomatic non-pregnant individuals over 15 years old (odds ratio (OR) 17.2 [13.8–21.6, 95% CI]) or asymptomatic children under 5 years of age (OR 3.8 [2.9–4.9, 95% CI]).

To obtain estimates of how acquisition of pregnancy-specific immunity influences detectability of infection with RDTs, we then used a previous model of the relationship between prevalence in the general population and cumulative exposure to MiP to account for likely patterns of prior exposure to infection during pregnancy by gravidity[17]. Our estimates suggest that the odds ratio for the pregnancy-related increase of detectability relative to non-pregnant adults, falls from 17.2 [13.8–21.6 95% CI]) in primigravidae to 4.05 [3.14–5.16, 95% CI] when a woman has experienced infection in one previous pregnancy and to 1.67 [1.22–2.34, 95% CI] if she has experienced infection in two previous pregnancies. By the fourth infected pregnancy, our estimates of sensitivity in pregnancy are no longer distinguishable from those outside of pregnancy (Fig. 2b).

We used multivariable logistic regression to find the best-fitting predictors of RDT sensitivity at subsequent ANC visits, using random effects intercepts to account for unknown or unmeasured factors between sites. This suggested gravidity remains a significant factor at later visits (OR 0.87 [0.78–0.96 95% CI] per additional previous pregnancy, $p = 0.005$), as does the presence of infection at the preceding visit (OR 0.70 [0.53–0.92 95% CI]). Other potential variables explored that were not kept within the best fitting model as measured by Akaike Information Criterion (AIC), after accounting for gravidity and PCR status, included a measure of parasite density of this previous infection (patent (RDT positive) or sub-patent), the number of times a woman had been previously tested, or the number of previous ANC visits she had attended. Despite the exploration of these factors, there remained substantial unexplained between-site variation within this best-fitting model ($p < 0.0001$).

In addition to these factors, RDT sensitivity in pregnancy is also likely to depend upon maternal age. This factor was not included in our analysis as sufficient granularity was not available for the relationship with RDT sensitivity outside of pregnancy[22]. Given the correlation between age and gravidity, it is likely that some age-dependent effects have been attributed to pregnancy-specific immunity. However, our results, which found that RDT sensitivity in primigravidae is substantially higher than would be expected in children (Fig. 2a), suggest the major determinants of the observed patterns are pregnancy-specific rather than age-specific.

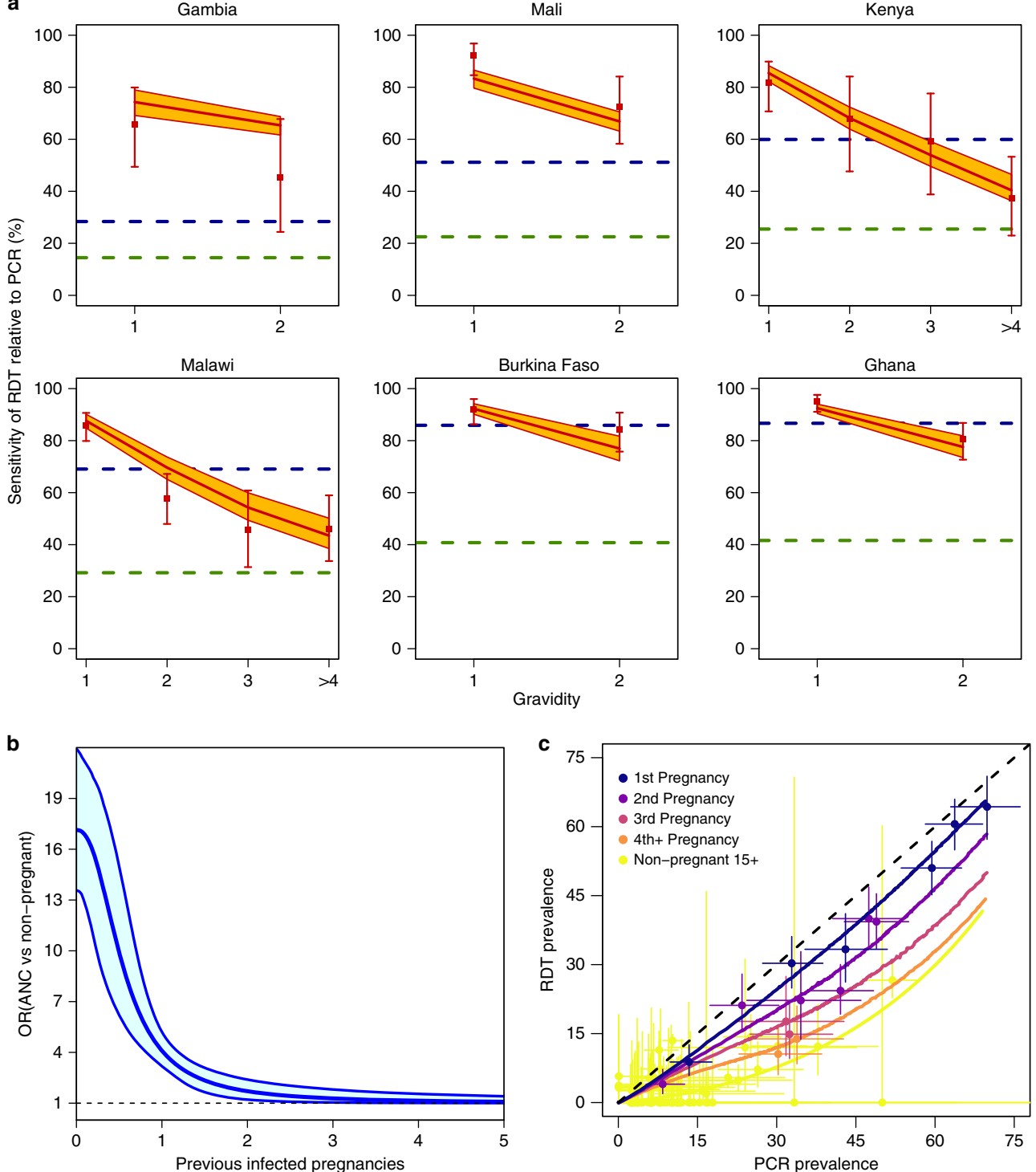

**Fig. 2 Comparing RDT performance at enrolment in ISTp trials to outside of pregnancy.** Figure shows **a** RDT sensitivity relative to PCR by gravidity across each of the six study countries (ordered by PCR prevalence in primigravidae), red dots and error bars show mean and 95% CI for RDT sensitivity within the trials. For comparison, the estimated sensitivity in non-pregnant individuals (male or female) aged >15 years old (green dashed line) and for children under 5 years old (blue dashed line) based upon the relationship between transmission and RDT sensitivity from Wu et al.[22] are also shown. The red line and orange polygon show the mean and 95% CIs for the best fitting model incorporating a declining boost in detectability of infection with RDT dependent upon the level of exposure in previous pregnancy, **b** shows the fitted relationship (see "Methods" section) from this model (blue line shows median and polygon 95% CI) of the odds ratio of detection at enrolment and non-pregnant individuals (male or female) aged >15 years old and **c** yellow dots and line show the data and the fitted relationship between PCR and RDT prevalence in non-pregnant individuals (male or female) aged >15 years old from Wu et al.[22]. Remaining colours show the estimated relationship between gravidity-specific PCR prevalence and RDT prevalence from this model (lines) compared to the data (dots with horizontal and vertical bars showing 95% CI for PCR prevalence and RDT prevalence, respectively).

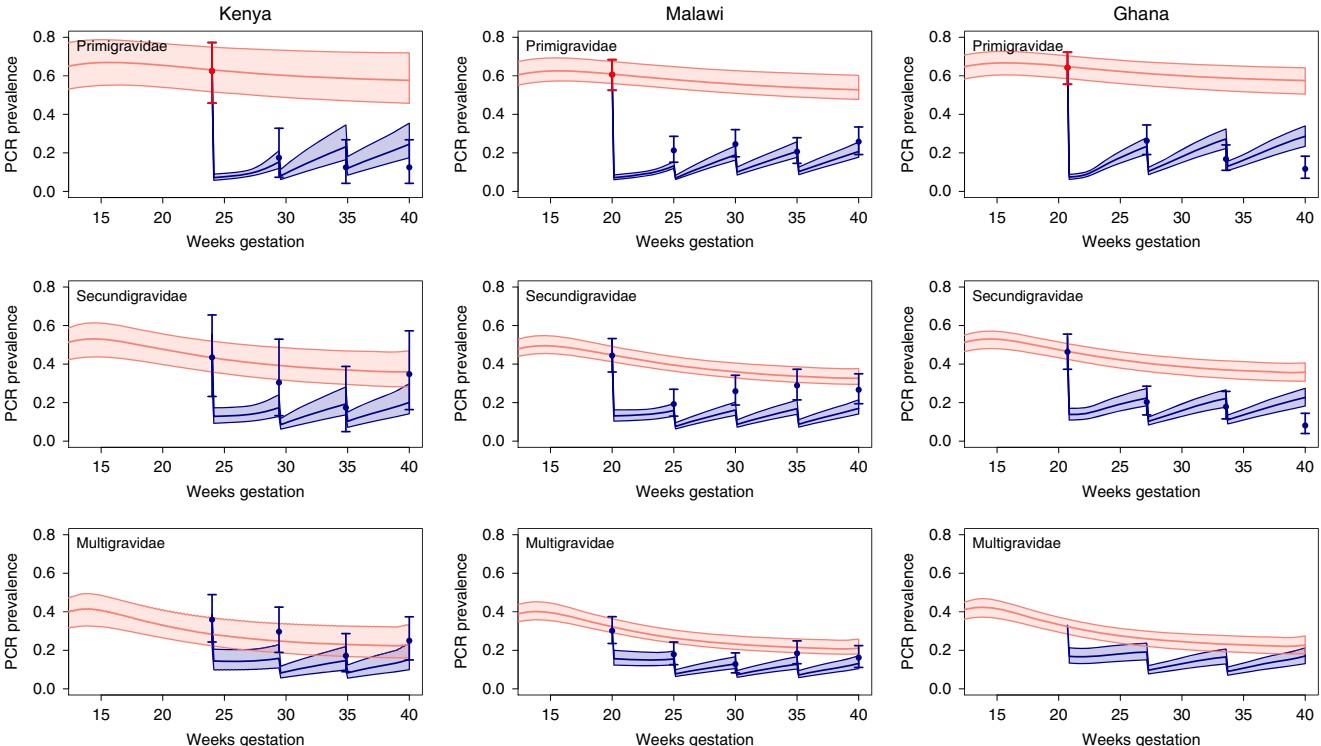

**Fig. 3 Comparison of simulated and observed dynamics of infection throughout trials of ISTp.** Figure shows observed PCR prevalence throughout successive screens during pregnancy (dots with error bars representing 95% CIs). Pink areas show the 95% CI for PCR prevalence throughout pregnancy in the absence of intervention. Red datapoints indicate observed prevalence at enrolment in primigravidae to which the model is calibrated for each trial, the remaining datapoints, marked in blue, represents dynamics the model aims to replicate, with sharp drops in prevalence corresponding to ISTp rounds. Blue areas show the 95% CIs generated by the posterior distribution of the fitted model in each scenario (see Supplementary Methods for full details) with blue lines representing the posterior median PCR prevalence. Note for the trial in Ghana only primi- and secundigravidae were recruited but the simulated output is still shown for completeness.

**Dynamics of infection throughout pregnancy with ISTp.** We updated a previous model of the relationship between malaria transmission and exposure to malaria infection throughout pregnancy to incorporate the factors described above[17]. We then assessed the extent to which this model replicated patterns of PCR prevalence throughout pregnancy within the three trials at each visit. This was done by restricting our analysis to women who received the modal number of screens in each trial (three screens prior to delivery in Kenya and Ghana, and four screens in Malawi). We also assumed that these dynamics could be approximated by simulations with an initial screen occurring at the median gestational age at which women could be enroled into each trial (24 weeks in Kenya, 20 weeks in Ghana and 20 weeks in Malawi for women receiving four screens), with subsequent visits spaced regularly until delivery at 40 weeks gestation.

We calibrated the model to match the observed PCR prevalence in primigravidae at enrolment in each trial, whereupon the model captured (Fig. 3) many of the observed dynamics of infection throughout the trials, namely:

i. decreases in PCR prevalence at enrolment by gravidity: driven in the model by reduced prevalence at conception due to the acquisition of non-pregnancy-specific immunity between pregnancies and improved clearance of parasitaemia during pregnancy related to pregnancy-specific immunity;

ii. a large decline in prevalence in primigravidae between first and second screens: explained in the model by the higher proportion of women testing positive by RDTs at first screen and associated treatment and post-treatment prophylactic effect;

iii. diminishing impact upon prevalence with increasing gravidity: driven by reduced detectibility of infection by gravidity due to pregnancy-specific immunity acquired in previous pregnancies;

iv. similar prevalence between tests from the second screen until delivery across all trials and gravidity categories: explained by the shorter window of exposure in which women can acquire new detectable infection (~20 weeks duration of gestation prior to first screen plus any residual infection acquired pre-conception versus ~4–8 gestation between screens) and the correspondingly smaller proportion of women benefiting from any post-treatment prophylactic effect from second screen onwards.

**Alternatives to IPTp-SP involving screening with standard RDTs.** To capture differential effectiveness of IPTp-SP as a function of the accumulation of parasite resistance mutations we defined three resistance scenarios (summarised in Fig. 4) as 'low prevalence quintuple' and 'high prevalence quintuple' SP resistance that map to those commonly observed in West and East Africa, along with a hypothetical scenario in which SP retains no antiparasitic activity, referred to as 'high sextuple' SP resistance areas. Within each scenario we compared six MiP prevention strategies: no intervention; IPTp-SP; ISTp; a hybrid approach (Hybrid-SSTp) wherein women are screened at first visit during the second trimester, provided an ACT if test-positive and SP if test-negative, then provided with IPTp-SP at subsequent visits; a second hybrid approach (Hybrid-ISTp) where women are tested at each ANC visit, provided an ACT whenever they test positive

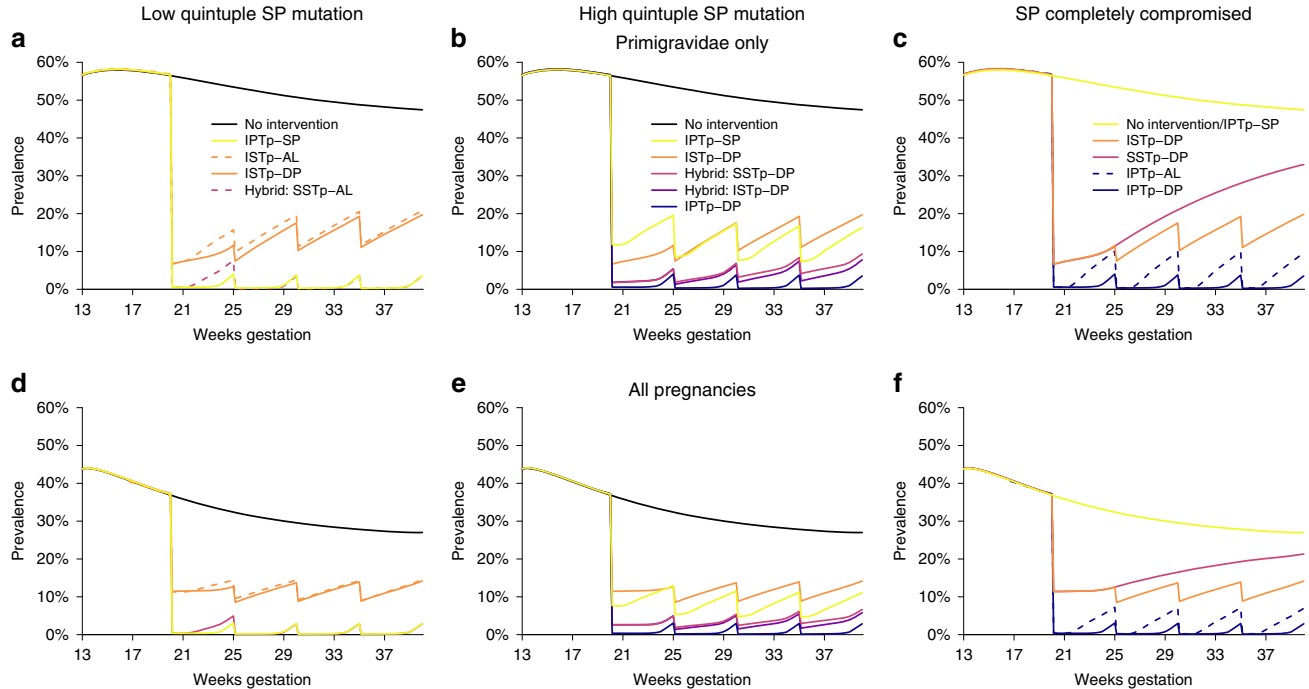

**Fig. 4 Simulated incremental benefit of alternative strategies to IPTp-SP by level of SP resistance.** Simulations are for high transmission settings (EIR = 100), top row shows peripheral PCR prevalence in primigravidae alone, bottom row averaged across all pregnant women. Left column, **a** and **d**, represent areas with low quintuple SP mutation, centre, **b** and **e**, with high quintuple mutation and right, **c** and **f**, represents a scenario with sextuple resistance where SP is assumed to no longer provide any protection. Simulations show the following strategies: no intervention (black lines), IPTp-SP (yellow lines), ISTp-DP (orange lines), Hybrid-SSTp (light purple lines), Hybrid-ISTp (dark purple lines) and IPTp-DP (blue lines). In general, for scenarios involving ACTs, simulations with DP are shown. In select situations simulations with shorter-acting AL are shown with dashed line. NB: In settings with low quintuple SP mutations, SP and DP are assumed to have equivalent efficacy so IPTp and hybrid strategies involving these drugs are indistinguishable when SP has no impact, ISTp and hybrid strategies using the same treatment drug are indistinguishable.

and SP otherwise (after the start of second trimester); and IPTp with an ACT. For each scenario involving ACTs we considered two possible drug combinations: artemether–lumefantrine (AL), which provides prophylaxis for around 10 days, and dihydroartemisinin–piperaquine (DP) which we assume provides prophylaxis of similar longevity to SP in the absence of resistance (see "Methods" and "Discussion" sections).

We compared scenarios in terms of two measures of exposure we consider likely to correspond to distinct sets of pathologies[1]:

- The proportion of women left with uncleared infection post-enrolment (Fig. 5), to capture impact upon pathologies associated with chronic placental infection such as intrauterine growth restriction and LBW. To incorporate the interaction between infection detectability using RDT and immunity, these estimates are then weighted by estimates of the number of LBW these infections would cause if left untreated (Fig. 5c).
- The risk of new infection later in pregnancy (Fig. 6), associated with a range of negative outcomes including preterm delivery, neonatal mortality and stillbirth[23,24].

Our results suggest that in low quintuple resistance areas, as in West Africa, where SP retains high efficacy and longevity, improving upon IPTp-SP strategies when delivered correctly, is likely to be challenging (Fig. 4a, d). The choice of ACT generally had limited impact upon prevalence when only provided to test-positive women (i.e. ISTp or hybrid strategies). However, hybrid strategies using AL resulted in higher infection prevalence than IPTp-SP between first and second visits, driven by the longer period of prophylaxis provided by SP[20] than AL[25] in areas of low quintuple resistance. This highlights the need to prioritise longer-

lasting ACTs when screening for infection at scheduled ANC visits in order to ensure women are provided with at least equivalent protection to IPTp-SP.

The ISTp trial in Malawi and Kenya were conducted in areas of high quintuple mutation SP resistance. Overall RDT sensitivity across all visits was 47% in these trials, meanwhile the risk that presumptive SP fails to clear existing infections is ~20% in these settings. Thus, it is unsurprising that our model incorporating these data suggests that IPTp-SP is more effective than ISTp in terms of cumulative exposure to infection during pregnancy measured by PCR (the 47% average risk of an untreated infection due to a false-negative RDT outweighs the ~20% risk of treatment failure with presumptive SP). However, accounting for higher sensitivity of RDTs earlier in pregnancy and in women with lower immunity, our results suggests that both IPTp-SP and ISTp have a large impact upon prevalence when compared to the counter-factual of no intervention (Fig. 4b, e). Moreover, our results suggest screening with RDTs early in the second trimester is effective at detecting the majority of early infections that would cause chronic intrauterine growth restriction leading to low birthweight if not treated (Fig. 5).

In Fig. 4b, c, e, f we show the effectiveness of IPTp with AL and DP. Neither drug is currently recommended for this purpose. Both are predicted to show incremental effectiveness in preventing infection over SP in areas where resistance has reached high levels of quintuple mutation or above. However, the incremental impact of DP, the focus of several ongoing studies, is substantially higher than the shorter-lasting AL. However, until a suitable, more effective, alternative drug to SP for IPTp has been recommended, our results suggest a hybrid strategy could be more effective than IPTp-SP alone in areas of high quintuple

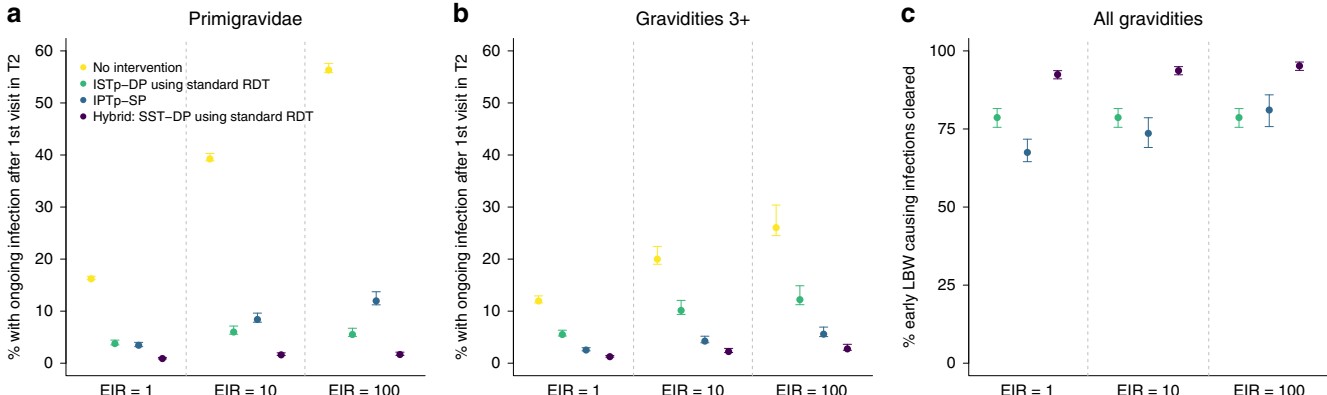

**Fig. 5 The relative effectiveness of IPTp, ISTp and hybrid strategies in clearing early infection. a** Shows the percentage of primigravidae with ongoing parasitaemia following their first visit in second trimester if they: receive no intervention (yellow dots), are screened with a standard RDT and treated with an ACT if positive (green dots), receive SP presumptively (light blue dots), are given an ACT if RDT+ and SP otherwise (hybrid strategy—dark blue dots). Error bars show 95% intervals based upon our uncertainty analysis for comparing the relative impact of intervention strategies (see "Methods" section), **b** Shows the equivalent figure but in women of gravidities 3 and above. **c** Shows the percentage of these early infections that would subsequently lead to LBW that are effectively treated based upon our modelled relationship between the detectability and severity of infection (NB: given these are ongoing infections this does not imply that treating these infections would necessarily avert all risk of LBW attributable to these infections—see "Methods" sction for full details).

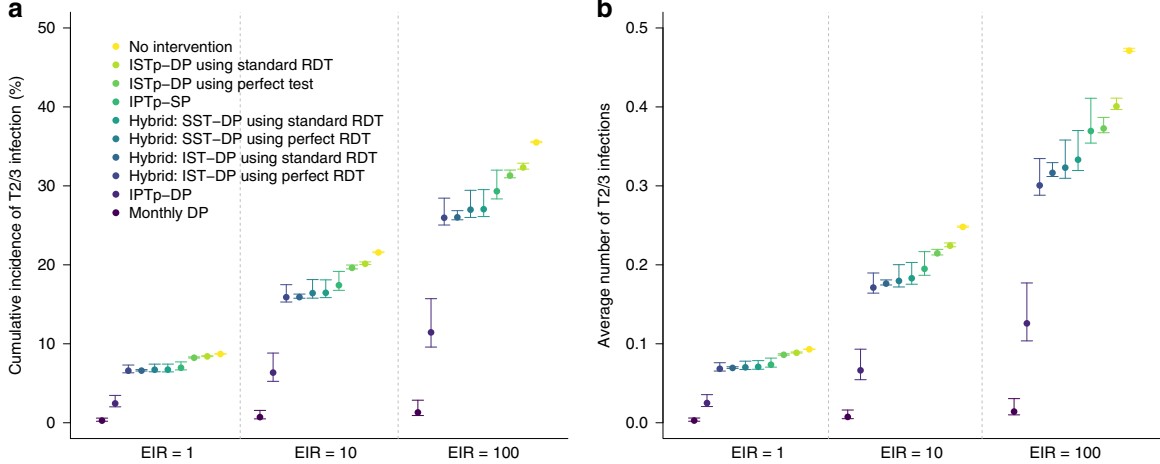

**Fig. 6 Impact of different strategies upon infection later in pregnancy in areas of high quintuple SP resistance.** Figure shows the impact of different simulated strategies upon the incidence of new (defined as either symptomatic or asymptomatic blood-stage) infection following a first ANC visit in the second trimester at 20 weeks gestation in areas of low, moderate and high transmission (EIRs of 1, 10 and 100). **a** Shows the percentage of women who will experience any new infection in the second or third trimester (T2/3), **b** shows the average number of new infections occurring throughout T2/3. Each strategy is assumed to involve three scheduled ANC visits occurring at 20, 27 and 34 weeks, except for "Monthly DP" (darkest blue) which involves five visits spaced 30 days apart from 20 weeks onwards. A perfect test refers to a hypothetical diagnostic with perfect sensitivity and specificity for peripheral or placental infection. Error bars show 95% intervals based upon our uncertainty analysis for comparing the relative impact of intervention strategies (see "Methods" section).

resistance or above. It ensures that RDT-positive infections, which are the higher-density, potentially more severe, infections are treated with a highly effective ACT, for which the curative efficacy is higher than for SP. Meanwhile, in contrast to ISTp strategies, women testing negative (both truly and falsely) still receive the same level of protection standard IPTp-SP (Fig. 5). Our model suggests that hybrid approaches at all scheduled IPTp visits, instead of just at the first visit, provides marginal incremental impact over the single screen-and-treat hybrid strategy (Figs. 4 and 6), whilst requiring substantially more resources due to repeated screening.

Our simulations also suggest that in areas with high quintuple mutant resistance, IPTp using a long-lasting drug, such as DP would be considerably more effective than IPTp-SP or any alternative strategies involving screen-and-treat strategies in

terms of their impact upon newly occurring infections from the second trimester onwards (i.e. following the timing when the first dose of IPTp would be scheduled to occur) (Fig. 6).

Given the limited data on the incremental sensitivity for infection in pregnancy of new highly sensitive RDTs (hs-RDTs)[26] we do not model the impact of specific hs-RDTs. Instead, we explored the extent to which more highly sensitive tests than standard RDTs in general could potentially improve the incremental impact of the strategies considered above.

For ISTp strategies, the added benefits of more sensitive RDTs may be small in high transmission areas if the bulk of adverse outcomes results from patent infections. Moreover, sub-patent infections missed by standard RDTs are more concentrated later in pregnancy when evidence for increased risk of adverse pregnancy outcome is not consistent (see the "Discussion"

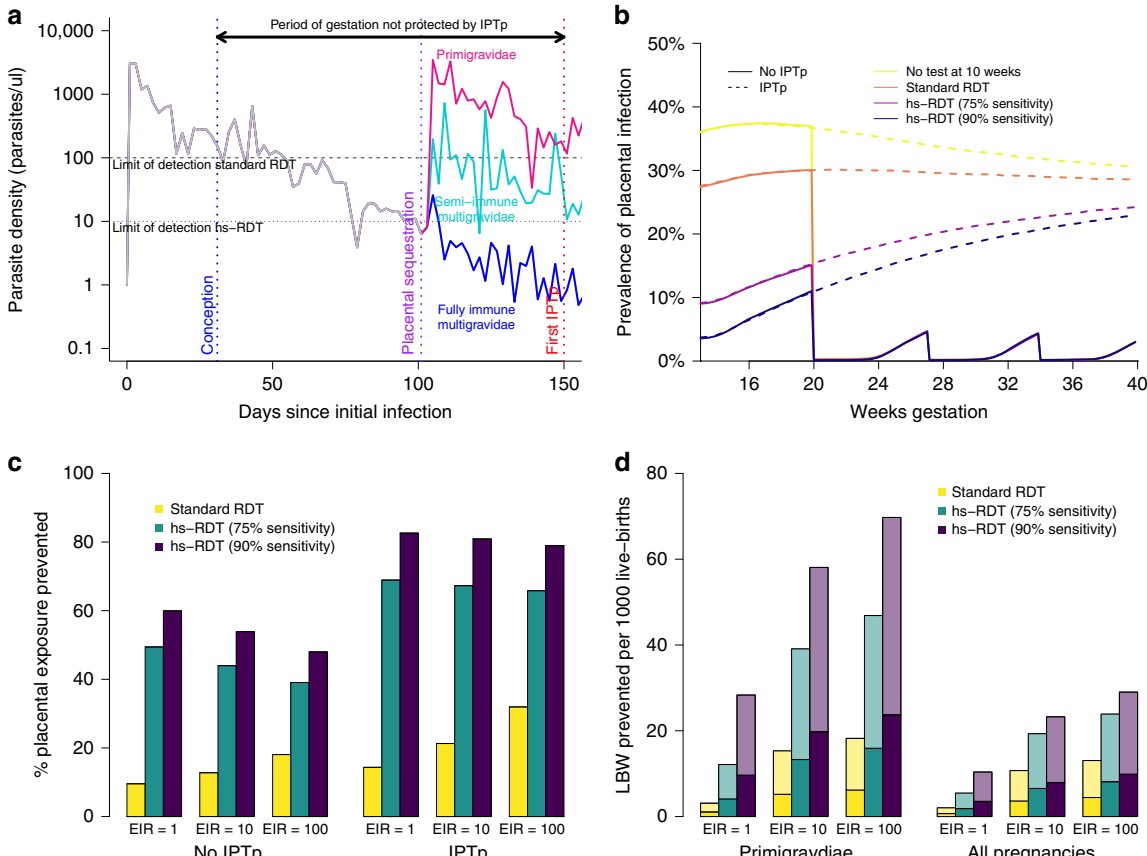

**Fig. 7 Potential impact of routinely testing for malaria during the first trimester. a** An illustration of the hypothesised mechanism by which the performance of standard RDTs are modified by gravidity. Women often experience chronic, asymptomatic parasitaemia outside of pregnancy which, as parasites are progressively cleared by the immune system, would eventually fall below the limit of detection of standard RDTs and be cleared if she had not conceived (grey line). If an asymptomatically infected woman becomes pregnant, and as her placenta develops so that maternal blood flows into the intervillous space (towards the end of the first trimester), the parasite undergoes antigenic switching, allowing it to bind to placental chondroitin sulfate A (CSA) receptors[1], multiplying to higher densities in women who have never experienced placental infection, leading to more severe and, due to higher concentrations of HRP2, more detectable infections (purple line). In subsequent pregnancies women mount a specific, acquired immune response, leading to better controlled, lower density and less detectable infection (turquoise and blue lines). **b** Shows a simulated example of the impact of testing and clearing infections at a first ANC visit at 10 weeks upon overall exposure to placental infection in primigravidae in a setting of EIR = 10 (ongoing peripheral infections are assumed to begin sequestering from the end of the first trimester onwards), simulations reflect our uncertainty in the sensitivity of the RDT at this time point, ranging from 26.8% (RDT sensitivity for asymptomatic infection in adults outside of pregnancy in such a setting based upon the relationship estimated in Wu et al.[22]) to 90% (the approximate sensitivity of standard RDTs at first visit in areas of high transmission in primigravidae in ISTp trials). **c** Shows the proportional impact this screening would have upon the mean duration of placental infection either in the presence or absence of IPTp-SP (assuming low SP resistance) and by transmission intensity. **d** The impact upon the risk of LBW according to our model relationship between the duration and stage of placental infection and LBW[5], two-thirds of these bars are coloured transparently emphasising our uncertainty in impact of IPTp-SP in terms of promoting catch-up growth[39].

section). Even in areas of high quintuple resistance, SP is likely to retain relatively high efficacy in clearing low-density infections missed by standard RDTs, resulting in >90% of infections being effectively cleared with a hybrid strategy using existing RDTs (Fig. 5). As a result, we estimate the incremental effectiveness of more sensitive diagnostics within hybrid strategies would be limited.

**Potential value of screening in the first trimester.** Although a large proportion of infections are likely to have been sub-patent at the beginning of pregnancy, by the time primigravidae receive the first dose of IPTp-SP, the density of infection has increased to the extent that very few remain below the limit of detection of standard RDTs (Fig. 7a)[18]. Clearing these infections during any first trimester ANC visit irrespective of the immediate density of the infection, is likely to have a large impact on the overall exposure to placental infection (Fig. 7c). Moreover (Fig. 7b), such

testing is predicted to have a large proportional impact on remaining exposure to placental infection in the presence of IPTp-SP (Fig. 7c), which leaves the first trimester entirely unprotected.

Our model suggests that a substantial number of infections acquired before or during the first trimester would lead to adverse outcomes if left untreated[5]. It also suggests that the impact of first trimester testing will depend strongly on gravidity, transmission, and the sensitivity of the test. The latter is likely to depend strongly upon poorly understood temporal dynamics of parasite replication in early pregnancy (Fig. 7a). Moreover, it is difficult to assess the extent to which future IPTp-SP will modify the impact of these early infections upon birth outcome.

## Discussion
By reanalysing malaria testing data from trials of ISTp we were able to generate the first quantitative estimates of the impact of

pregnancy upon the detectability of infection using RDT. These relationships provide more nuanced understanding as to the failure of ISTp to show incremental effectiveness compared to IPTp-SP in trials Our estimates suggest that infections missed by standard RDTs lead to a greater proportion of inadequately treated infected women than providing SP presumptively (i.e. the negative effects of misdiagnosed infections outweigh those of treatment and prophylaxis failures). However, in these settings our simulations suggest that both IPTp-SP and ISTp, whilst failing to provide optimal protection, effectively prevent the majority of infections when compared to women without any intervention.

This finding, that ISTp has substantial intrinsic impact relative to no intervention despite not being superior to IPTp, is supported by a recent meta-analysis of four trials comparing these two strategies: when pooled these studies show that babies born in the IPTp-SP arms had a 25 g higher mean birthweight than in the ISTp arm (95% CI 7–44, $p = 0.0088$, $I^2$ 0%, 8659 pregnancies)[11]. In absolute terms, this difference is small compared to the 79 g (95% CI 13–145) seen with IPTp-SP when compared to placebo or passive case detection. Consequently, ISTp, has potential advantages over current practice in some countries that do not deploy IPTp due to concerns about SP efficacy, or, if an adequate replacement drug for SP within IPTp regimens cannot be identified, if SP became completely ineffective due to further development of resistance in the future (Fig. 4). This might be the case if the 'sextuple' A581G resistance mutation, established in specific foci in East Africa, becomes more prevalent and widespread.

There is not sufficient data to specifically include recently developed highly sensitive tests for HRP2 within our analysis. However, our results suggest more highly sensitive diagnostics in general could improve ISTp strategies, and if sufficiently sensitive, could provide incremental effectiveness over IPTp-SP in terms of parasitological outcomes, such as infection prevalence by PCR at delivery in areas of high quintuple mutation SP resistance. However, the clinical implications of any increased effectiveness to detect low-density infections, which are more common in multigravidae and later in pregnancy remain to be determined. The association between low-density infection in the second trimester onwards and pregnancy outcome is not consistent[27–31]. However, as transmission falls, the density of peripheral and placental infection at delivery in multigravidae increases[4], presumably reflecting a lower level of exposure to malaria during previous pregnancies. Of all trial sites, RDT sensitivity at enrolment in primigravidae was lowest in the Gambia, the trial site with the lowest transmission. This may reflect lower density of infection prior to placental development, as the sensitivity of infection by RDT outside of pregnancy falls as transmission declines[22,32]. In these areas, more sensitive RDTs could substantially improve the ability of ISTp to detect and treat what would otherwise be long-lasting infections in women lacking pregnancy-specific antimalarial immunity. More data are required from studies measuring RDT sensitivity in pregnancy in areas of low transmission in order to assess this hypothesis.

In areas with high prevalence of quintuple SP resistance, hybrid strategies show promise as a solution to offset the respective weaknesses of IPTp-SP and ISTp. A potential advantage of hybrid strategies over IPTp is that they prioritise the use of highly effective ACTs to those with the higher density infections early in pregnancy most likely to cause harm. Retaining IPTp-SP for women who test negative still receive the current standard of care and ensures that women with low-density sub-patent infections are not left untreated. Standard RDTs perform well at the first antenatal visit in the second trimester, when prevalence and parasite densities are highest, largely offsetting the need for more sensitive diagnostics, at least whilst SP retains the majority of its

curative efficacy (which is the case even in high quintuple resistant areas[3,20]). However, given the complexity of multi-day ACT dosing regimens, the theoretical advantages (in terms of efficacy with 100% adherence) and real-life advantages (accounting for adherence), need to be carefully considered to ensure that switching strategies does not lead to lower protection relative to IPTp-SP in practice.

Hybrid strategies may only represent an interim solution if SP resistance continues to increase and SP effectiveness progressively declines[19]. Adding screening at first IPT-SP visit only alleviates some of the risk associated with these infections, and more effective chemoprevention with longer-lasting drugs, such as DP is likely required to provide larger incremental benefits to pregnancy outcome[9,11,33,34]. Two confirmatory trials of IPTp-DP are currently ongoing in Kenya, Malawi and Tanzania (clinicaltrials. gov NCT03208179 and NCT03009526). Estimates of SP prophylactic longevity outside of pregnancy suggest equivalent prophylactic longevity to DP (approximately 1 month)[25] in areas where the quintuple SP resistance mutation is large. However, there remains a dearth of data allowing the direct comparison of the effectiveness of the two drugs in pregnancy in such settings, limiting our ability to provide guidance on the relative merits of IPTp when SP resistance is low. Such data could also help to provide insight into the extent to which SP has impact upon non-malarial causes of adverse pregnancy outcome which we do not capture in our analysis.

Our analysis highlights that a high proportion of pregnant women are already infected prior to the second trimester, the earliest stage at which doses of IPTp-SP can be initiated. A large proportion of these infections are likely to have been acquired early in pregnancy or prior to conception, as evidenced by genotyping pre-conception infecting parasites in Benin[18] and the high prevalence of infection in women first attending ANC outside of the transmission season in seasonal settings[35]. Our results support the findings from Benin in suggesting that in primigravidae, low-density infections at conception persist, multiply and sequester within the placenta at crucial stages of development[18]. Thus, adding screening for malaria in the first trimester could have important benefits. This relies upon women being aware of their pregnancy and ANC provision and attendance during this period, though, first trimester ANC is a strong focus of updated 2016 WHO ANC guidelines[36], which now recommends a first ANC visit prior to 12 weeks gestation, and the drive to improve ANC as part of the wider sustainable development goals[37].

Estimating the impact of treating first trimester infections upon birth outcome, and the extent to which this depends upon subsequent IPTp uptake, is challenging as most studies measuring associations between early infection and birth outcome do so in the context that these infections are effectively treated upon detection. Some adverse pregnancy outcomes associated with first trimester infection, such as disruption of the development of aspects of placental vasculature, may be irreversible[38], whereas for others, e.g. intrauterine growth restriction, IPTp-SP may allow recovery and catch-up growth later in pregnancy[39]. In the absence of randomised controlled trials of the impact of first-trimester screening, the findings that parasite densities are likely to be on the rise early in pregnancy, and the increasing data suggesting a negative impact of these infections upon placental and foetal development, even in the presence of IPTp-SP[15,38], suggest there is no threshold level of parasitaemia under which women can be safely exposed during the first trimester. Providing presumptive antimalarial treatment or prophylaxis at this stage of pregnancy is challenging as ACTs are only recommended for case-management in the first trimester. However, the ability to identify women carrying infections at this stage by testing with a

highly sensitive diagnostic test, and thus treat infections before they have the chance to multiply and sequester within the placenta, has the potential to provide substantial and lasting benefits to maternal, foetal, neonatal, and infant health. The only published study assessing the performance of existing next-generation highly sensitive RDTs during pregnancy detected a statistically insignificant higher number of PCR-positive infections than conventional RDTs[40]. However, this study was conducted in an area of low transmission, with testing conducted throughout gestation and at delivery. Interpreting these results in terms of the value of such tests for first trimester screening in areas of higher transmission is challenging.

The extent to which hybrid strategies would affect uptake of ANC-based interventions aimed towards preventing MiP (IPTp and ITNs) is unknown and will be a large determinant of the impact of such a shift in policy. Since adopting a single screen and treat hybrid approach as policy, the uptake of routine testing as an ANC-based intervention in Tanzania has been rapidly increasing, from 36.7% in 2014 to 88.8% in 2017[14]. This uptake is particularly impressive in the history of scale-up of IPTp, both in Tanzania, where IPTp-SP became policy in 2001 but where only 56% of pregnant women received two or more doses of SP in 2017, and more generally across Africa[6]. Understanding whether this rate of uptake would be mirrored in other countries, and whether it leads to a higher proportion of ANC attendees receiving any malaria-specific intervention, will be key to understanding the overall role testing may have in improving the limited protection from malaria currently provided to pregnant women.

This study has several limitations. The epidemiology of MiP is complex, particularly placental infection, which can only be reliably measured at delivery. As a result, and given the challenges associated with quantifying the attributable burden of multi-factorial negative pregnancy outcomes, such as LBW, preterm delivery, and foetal loss, we were not able to include direct estimates of the impact of these interventions upon many of the negative effects of malaria in pregnancy. Although we can estimate the impact of different strategies on the incidence of new infection, we could not quantify these effects on the burden of clinical malaria, neither could we quantify impact on prematurity and stillbirth, which are likely to depend upon timing during gestation and transmission intensity[23]. Our analysis does not include any consideration of optimal strategies to protect HIV-infected pregnant women who currently receive daily cotrimoxazole, which provides sub-optimal protection from malaria[41]. Finally, we do not capture the potential value data from ANC-based screening to improve malaria surveillance[42].

In conclusion, our modelling suggests that screening and treatment with the current generation of RDTs would not provide incremental effectiveness relative to WHO's existing IPTp-SP strategy, even in areas with high quintuple mutation SP resistance. However, screen-and-treat strategies may have incremental benefit if the effectiveness of IPTp-SP is reduced further by resistance, especially in areas with high prevalence of sextuple SP mutants. Our model suggests that hybrid strategies integrating screening at the first antenatal visit into existing IPTp-SP regimens are potentially beneficial in areas with high prevalence quintuple mutation SP resistance. Moreover, screening women routinely for malaria in the first trimester and providing effective treatment could provide substantial benefit, particularly if suitable highly sensitive diagnostics for first trimester infection can be identified.

## Methods

### Estimating the effects of pregnancy on RDT performance within ANC.
We related the observed sensitivity of RDTs at enrolment in the ISTp trials to RDT sensitivity in the general population and the acquisition of pregnancy-specific

immunity due to prior exposure to MiP according to the following function (see Supplementary Methods for full details of models and model fitting):

$$\text{Odds}\left(S_{ij}^{W}\right) = \text{Odds}\left(S^{A}\left(x_j\right)\right)\left(1 + \frac{\beta}{\left(1 + y_{ij}/\delta\right)^{\nu}}\right). \quad (1)$$

Here $S^{A}(x_j)$—describes the probability, $p$, that a PCR-positive infection in the general population is detected by RDT, following a function of overall PCR prevalence within the setting, $x_j$. Odds are related to probability $p$ by the general relationship $\text{Odds}(p) = \frac{p}{1-p}$.

The sensitivity $S_{ij}^{W}$ represents the probability that a PCR-positive infection of a newly enroled pregnant woman $i$ within site $j$ is detected by RDT. The odds of detection in primigravidae are boosted by a constant $\beta$ relative to the equivalent odds for the probability of detecting infection by RDT outside of pregnancy. This pregnancy-related boost in detectability, relative to adults in the general population, then decreases with increasing number of pregnancies in which a woman has previously been exposed to malaria, denoted $y_{ij}$. This follows a Hill function with offset parameter $\delta$ and power parameter $\nu$.

Neither the sensitivity of RDTs outside of pregnancy nor the exposure history in previous pregnancies were available in the data. Instead, we relied on the following fitted relationship between RDT sensitivity and PCR prevalence obtained by Wu et al.[22]

$$S^{A}\left(x_j\right) = \left[x_j\left(1 + \exp\left(-\left(\mu_A * \text{Odds}\left(x_j\right) + \sigma_A\right)\right)\right)\right]^{-1}, \quad (2)$$

where $\mu_A = 1.30$ and $\sigma_A = -1.38$ are the best fitting parameters obtained by fitting this model to matched cross-sectional RDT and PCR samples in people aged over 15 (parameters obtained from fitting to matched RDT and PCR data from children under 5, young adults aged 5–15 and all-age surveys were also included within separate model fits for comparison)[22]. Working within a Bayesian framework we were then able to simultaneously fit this model and a previously developed model of the relationship between malaria transmission and exposure to MiP[17] (see Supplementary Methods for full details) to the gravidity-specific patterns of RDT and PCR detection across each setting, accounting for uncertainty in $y_{ij}$, the number of previous pregnancies during which each woman would have been exposed to malaria. This provided inference on the parameters $\beta$, $\delta$ and $\nu$ determining the impact of pregnancy upon detectability of infection using RDT. This model was fitted alongside models where the sensitivity of RDTs at enrolment were independent of either transmission intensity or gravidity, or independent of both, and compared using the deviance information criterion (DIC) (see Supplementary Methods for full details of model fitting).

The probability of detecting infection at later ISTp visits was modelled as a separate logistic multivariable regression accounting for random effects between study sites. Gravidity, infection status of the previous test and overall throughout pregnancy, and the number of previous visits or tests were all included as potential predictors of RDT sensitivity. Model selection was carried out using AIC, and parameters of the best-fitting regression were included in the dynamical model (see Supplementary Methods for a detailed description of this analysis).

### Modelling the impact of interventions upon parasite prevalence.
We extended our existing model, linking transmission in the general population to the risk and burden of MiP and effectiveness of IPTp[3] to incorporate our estimates of RDT sensitivity at enrolment and throughout pregnancy (see Supplementary Methods for full details). The overall fit of this model was assessed by visually comparing PCR prevalence throughout pregnancy in the ISTp arm in each trial site, with data restricted to women with the modal number of visits in each site, with 95% uncertainty intervals of trajectories of PCR prevalence throughout pregnancy generated by the model using 1000 draws from the joint posterior distribution of $\beta$, $\delta$ and $\nu$ and from the parameters within the final regression model of RDT sensitivity after enrolment, with the model calibrated by varying EIR to match a draw from the 95% confidence interval PCR prevalence at enrolment in primigravidae in each site (see Fig. 3).

For all protocols involving the use of SP we considered three separate scenarios with respect to the resistance of $Pf$ to the drug on the basis of the prevalence of the quintuple K540E resistance mutation and in vivo efficacy data: 'Low quintuple resistance' (K540E mutation prevalence <15%), women are protected with almost no treatment failures over the period of a month (here we assume a Weibull-distributed period of protection which SP prevents over 50% of infections until mean period of prophylaxis of 28 days, reflecting the observation that reinfections following treatment in these areas appear to begin occurring around this duration post-treatment), 'high quintuple resistance' (K540E mutation prevalence >85%), where the risk of infection recrudescence has been estimated to be 21.6% but re-infections appear to occur readily around a week after treatment, and an 'intermediate quintuple resistance' category (15% < K540E prevalence <85%) where treatment fails to clear infection around 10% of the time and prophylaxis appears to last around 2 weeks[20]. There remains no efficacy data on the effects of the A581G sextuple mutation in pregnant women, but there is evidence to suggest that IPTp-SP efficacy may be severely compromised in settings with prevalence >37%[19]. As a result, we also carried out simulations under the scenario that SP is provided but has no impact. In the absence of data, we assume that treatment failure occurs randomly with respect to gravidity, gestational time, or

whether an infection is detectable by RDT. The ACTs AL and DP were also assumed to have near perfect efficacy in clearing ongoing parasitaemia. AL was assumed to have a mean prophylactic half-life of 14, matching that estimated outside of pregnancy by Okell et al. (13.8 days [range 10.2–22.8 days])[25]. In the same analysis Okell et al. estimated a prophylactic half-life of DP outside of pregnancy of 29.4 days [range 16.4–48.8 days][25], similar to our assumed duration of effectiveness of SP in areas of low quintuple resistance. As a result, in the absence of specific data comparing duration of effectiveness of SP and DP in pregnancy in areas of low quintuple resistance, we assumed the same prophylactic profile for both drugs and avoid drawing conclusions as to the relative merits of the two combinations in such settings.

Estimates of the extent to which screening infection in the first trimester will prevent low-birthweight are based upon a previous analysis looking at different models of the relationship between exposure to malaria and malaria-attributable LBWs, the best fitting of which involved a relationship depending upon the level of chronic placental infection during pregnancy which was modified by exposure to infection during previous pregnancy[5]. We make the conservative assumption that, prior to the beginning of the second trimester, variation in detectability of infection using RDT will be random with respect to gravidity. When estimating the proportion of LBW-causing infections detected by standard RDTs at first visit in the second trimester (e.g. Fig. 5), we incorporate the dependence between both LBW risk and RDT sensitivity and pregnancy-specific immunity (see Supplementary Methods for full details and parameter values of our model of malaria-attributable LBW). However, we again make a likely conservative assumption with respect to any advantage of RDT-based screening that, for a given level of pregnancy-specific immunity, there is no difference between RDT detectable and undetectable infection in terms of attributable LBW risk. As highlighted in the "Results" section, we are not able to estimate the potential impact that clearing infection later in pregnancy through IPTp-SP may have upon this risk.

WHO recommends IPTp is given at 13 weeks gestation then subsequently every 4 weeks[36], however, to avoid presenting an over-optimised picture of interventions in pregnancy, we here model IPTp (and corresponding ISTp or hybrid) schedules of three or four contacts rather than monthly, which are more reflective of the number of ANC contacts that women generally have across Africa[3].

**Reporting summary**. Further information on research design is available in the Nature Research Reporting Summary linked to this article.

## Data availability

There are three separate primary data sources used in this analysis:

1) Matched RDT and PCR samples from a trial of Intermittent Screening and Treatment in pregnancy based in four countries in West Africa (see ref. [8] in the main manuscript for full details).

2) Matched RDT and PCR samples from a trial of Intermittent Screening and Treatment in pregnancy in Western Kenya (see ref. [9] in the main manuscript for full details).

3) Matched RDT and PCR samples from a trial of Intermittent Screening and Treatment in pregnancy in Malawi taken from reference (see ref. [10] in the main manuscript for full details).

4) A review of matched RDT and PCR prevalence in non-pregnant adults (see ref. [22] in the main manuscript for full details).

Figures 1–3 show data from sources 1–3. Supplementary Fig. 1 and Supplementary Tables 2–4 show output from fitting to these data. Figure 2c show data from source 4.

Source 1 is available subject to agreement with the original authors from the LSHTM Data Compass https://datacompass.lshtm.ac.uk/4/

Sources 2 and 3 are available for access with the WorldWide Antimalarial Resistance Network (WWARN) at www.WWARN.org. Requests for access will be reviewed by a Data Access Committee to ensure that use of data protects patient privacy according to the terms of consent and ethics approval.

Source 4 is freely available to download from a supplementary data file from https://www.nature.com/articles/nature16039.

## Code availability

Source code of the mathematical model developed and used within this analysis, along with a compiled version and compilation and running instructions are available open access at the following repository: www.github.com/patrickgtwalker/malaria_in_pregnancy_istp_model_open.

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

## Acknowledgements

P.G.T.W., K.K., J.H., S.K. and F.O.t.K. acknowledge funding support from EDCTP as part of the EDCTP2 programme supported by the European Union (grant number CSA-MI-2014-276 IMPPACT). J.H. and F.O.t.K. acknowledge funding support from WWARN (which is funded by the Bill & Melinda Gates Foundation; grant number OPP1181807). P.G.T.W. and A.C.H.G. were supported by the MRC Centre for Global Infectious Disease Analysis (MR/R015600/1): This award is jointly funded by the UK Medical Research Council (MRC) and the UK Department for International Development (DFID) under the MRC/DFID Concordat agreement and is also part of the EDCTP2 programme supported by the European Union. The findings and conclusions in this report are those of the author(s) and do not necessarily represent the official position of the Centers for Disease Control and Prevention.

## Author contributions

P.G.T.W., M.C., A.C.H.G., and F.O.t.K. conceived and designed the study. P.G.T.W., M.C., J.G., H.S. conducted analyses and prepared figures. M.C., J.G., K.K., J.E.W., S.O.C., C.K., S.T., S.R.M., J.H., V.M., L.K.-P., K.B., S.K., H.T., M.M., M.D., and F.O.t.K. contributed data or aided with the interpretation of data. All authors contributed to drafting and revising the manuscript.

## Competing interests

The authors declare no competing interests.
