## [Peer Review File · Nature Communications]

Reviewers' comments:

Reviewer #1 (Remarks to the Author):

This is quite a long and dense paper which uses modelling approaches to investigate areas of current interest in relation to the optimal approaches to prevention of malaria in pregnant women and its consequences for the baby. It is focused on Africa, where presently WHO recommends regular doses of sulphadoxine pyrimethamine administered at antenatal clinical to all women in second and third trimesters, irrespective of malaria infection status, termed IPTp-SP. This policy faces several challenges, most notably drug resistance to SP (which divides into medium and high-level resistance), the relative dearth of data on alternate drugs for use in IPTp, and uncertainty regarding both the consequences of, and best strategies for, malaria prevention in first trimester. The project includes modeling of the potential uses of existing and potential new point of care tests (RDTs) to help minimise the impact of the disease, including impacts in first pregnancy (when malaria risk is highest) and later pregnancies (in which acquisition of immunity reduces infection severity). Finally, it also attempts to determine how much of the large burden of low birth weight found in malaria-endemic Africa might be ameliorated by some of these different interventions. These are all timely and important questions.

Major comments

1. While early figures in the paper have 95% CIs for some of the models provided, in later figures these are not given, yet as the paper proceeds, the complexity of the data (and thus presumably the margins of error in the estimates) increases. Figures 5 and 6 would seem to be the least precise yet there are no error bars or indications of the confidence around these estimates.
2. In Figure 2 a, the derivation of the green dotted lines (sensitivity in children U 5) does not seem to be in keeping with either expected sensitivity of RDTs in this age group or with the data these were apparently derived from. In Wu fig 4, the graphs are not in keeping with these lines and the text state "for children under 5 years of age, RDTs detected 81% (95% CI = 74–89%) of PCR-positive infections"- while this is lower at low transmission, the data seem to be incorrect. Moreover the sensitivity of RDTs relative to PCR is HIGHER in the young vs older children in the source paper. It may simply be the legends are switched for the lines, but this needs careful confirmation. Text in lines 181-3 needs to be revised. Are there any implications of this apparent mistake for later model calculations? I would expect the U5 and primigravid sensitivities to be about the same based on other approaches to these populations, and their relative immunity.
3. Line 199-201: I do not follow how differences in non-pregnancy specific immunity contribute to the decrease in PCR prevalence by immunity at enrolment (I know of no evidence that this is gravidity dependent). The better explanation is immunological memory related to pregnancy specific immunity, which has potentially been boosted by new exposures from 12- 20 or 24 weeks.
4. Line 238: what is the basis for the statement that DP and SP offer similar post-treatment prophylaxis? It does not appear apparent for the references, and the combination of the longer half life of piperazine and the prevalence of low grade markers of SP resistance even in "sensitive" areas (which affect MIC and this prophylaxis, White N PloS Med 2005) makes these drugs less likely to be equivalent that the authors assume.
5. Figure 4 – IPTp with AL might help make a point in the simulations, but it is not a real world alternative strategy for IPTp and it is debatable whether it should be included. If it is, the text needs to indicate it is simply there to illustrate how drug half life affects the models.
6. Line 319 onwards: we may not have enough data to model the effects of hs-RDTs, as the authors point out the tiny studies to date do not show these tests are much better than standard RDTs in pregnancy, so trying to develop models which take this into account may indicate they have few advantages, while models that assume they have a 10 fold lower limit of data will not fit with the "lived experience" of such tests to data.
7. Line 399 onwards: the "intrinsic value" of IST is not really established – it is a lot more expensive than IPTp with SP and the countries where it might be deployed are presumably low transmission areas- while much of the modelling presented focuses on higher transmission areas. The data seem to illustrate why IST has not been endorsed by WHO rather than making a case for its being considered. In the case of complete failure of SP, DP IPTp would be a clearly better alternative as several figures show.

Minor comments

1. Line 171 is AIC defined somewhere?
2. Line 208 tretment for treatment
3. Line 296-8 the sentence is incomplete.
4. Figure 5- the title is not truly reflective of the message. Looking at Fig 5 a one would think that no drug would fail to clear 15% of infections at low EIR. Different wording might better reflect the message here.
5. Line 580-1- see other comments about equivalence (or not) of prophylaxis from DP and SP.
6. References 8, 11, 14, 15 have incomplete citations
7. References 6 and 41 are duplicates.
8. Supplementary material P 6 placental infections: line 5-7. Replace with "... expressing VAR2CSA, the Plasmodium falciparum erythrocyte membrane protein 1 variant which binds to chondroitin sulphate A"
9. Same page point 1) sequestered, not sequested
10. Page 8: I am unclear what is meant by "rate of progression through past stage infection". As far as we know this is an end stage event- it does not resolve.
11. Page 11 sentence "We therefore developed as model of RDT sensitivity at subsequent visits as a using logistic regression." needs editing
12. Paragraph starting "Defining"- sentence meaning unclear.

Reviewer #2 (Remarks to the Author):

The data analyses appear to have been carried out very competently.

A major issue that is barely addressed is the practical difficulties of implementing the different strategies in the challenging setting of MCH clinics. Diagnostic testing and/or differentiated drug regimes for different stages of pregnancy or parity represent significant complications to clinic operations. The paper does not engage with the question of the likely compliance and adherence to alternative drug regimes until line 438. Similarly, the section beginning line 344 does not address the issue that many women will not know that they are pregnant during the first trimester.

(Figure 2b shows fitted values of an odds ratio, but it is not explicit in the legend what is the outcome being modelled).

Reviewer #3 (Remarks to the Author):

This is an important and timely topic given a number of new trials, changing epidemiology of malaria, and dynamic patterns of SP resistance.

The case for relating more easily measured endpoints (e.g. parasite prevalence + density) to impact through modeling is compelling described in the introduction (L87-95).

This is especially useful in the handling of prospective scenarios around treatment with ACT in early pregnancy (L344 + Fig 7a/b).

L123 + Fig. 1: An earlier statement of how enrollment worked in these trials would be helpful for context. Later, on L193, this is clarified in the introduction of the modeling work (Fig. 3)

L139-141: Again, the reader first encounters some odds ratios that would benefit from an earlier and more detailed description of how the possible differences in age distributions of enrolled primigravidae compare to the general 15+ population, and the expected scale of exposure- or immunity-related impact that could have (introduced at L178-180). The young age-group result is compelling, but one wonders whether -- in conjunction with "unexplained between-site variation" (L175) -- a more complete picture can be told.

Fig 2a: The varying gravidity axes by panel was a bit confusing, but I suppose there's an aesthetic trade-off with leaving a bunch of white space in 4 panels if they're all the same scale.

L175 + Fig 2c: What role does the seasonality of enrollment of the gravid cohorts play in shifting the RDT/PCR relationship across sites? That might be relevant in Mali and Burkina -- both in projecting the results + in the operational opportunities available to different intersections of gestation (9 months) and high-season (3-4 months).

L392: Is there a typo in this sentence with the acronyms?

Reviewer #1 (Remarks to the Author):

This is quite a long and dense paper which uses modelling approaches to investigate areas of current interest in relation to the optimal approaches to prevention of malaria in pregnant women and its consequences for the baby. It is focused on Africa, where presently WHO recommends regular doses of sulphadoxine pyrimethamine administered at antenatal clinical to all women in second and third trimesters, irrespective of malaria infection status, termed IPTp-SP. This policy faces several challenges, most notably drug resistance to SP (which divides into medium and high-level resistance), the relative dearth of data on alternate drugs for use in IPTp, and uncertainty regarding both the consequences of, and best strategies for, malaria prevention in first trimester. The project includes modeling of the potential uses of existing and potential new point of care tests (RDTs) to help minimise the impact of the disease, including impacts in first pregnancy (when malaria risk is highest) and later pregnancies (in which acquisition of immunity reduces infection severity). Finally, it also attempts to determine how much of the large burden of low birth weight found in malaria-endemic Africa might be ameliorated by some of these different interventions. These are all timely and important questions.

We thank the reviewer for their positive feedback.

Major comments

1. While early figures in the paper have 95% CIs for some of the models provided, in later figures these are not given, yet as the paper proceeds, the complexity of the data (and thus presumably the margins of error in the estimates) increases. Figures 5 and 6 would seem to be the least precise yet there are no error bars or indications of the confidence around these estimates.

We agree with the reviewer that uncertainty in these figures needed to be more comprehensively addressed whilst noting that adequately conveying uncertainty is challenging when comparing the effects of multiple interventions across multiple settings and gravidity categories. This is especially the case as the mean effect size of a difference in intervention is not particularly well-correlated with whether a difference is positive or negative - a topic that has also been the subject of much discussion between co-authors. For example, in figure 4, using a shorter-lasting drug (e.g. AL) within a screen-treat strategy does not lead to a large increase in prevalence throughout gestation but it is highly certain that there would be such an increase both in intuitive terms and within the model - differences such as these would be obscured if we were to present a full range of uncertainty including factors such as uncertainty in timing of ANC visits or the relationship between transmission and exposure to malaria outside of pregnancy (i.e. uncertainty in the non-pregnancy-specific part of the model).

As a result, for these figures we now present error bars that reflect our uncertainty in the main drivers of differential intervention impact within the model – namely our uncertainty in the sensitivity of the RDT and how this relates to pregnancy-specific immunity and the efficacy and prophylactic profiles of each drug. This has been described in detail in the methods and supplementary information.

2. In Figure 2 a, the derivation of the green dotted lines (sensitivity in children U 5) does not seem to be in keeping with either expected sensitivity of RDTs in this age group or with the data these were apparently derived from. In Wu fig 4, the graphs are not in keeping with these lines and the text state “for children under 5 years of age, RDTs detected 81% (95% CI = 74–89%) of PCR-positive infections” - while this is lower at low transmission, the data seem to be incorrect . Moreover the sensitivity of RDTs relative to PCR is HIGHER in the young vs older children in the source paper. It may simply be the legends are switched for the lines, but this needs careful confirmation. Text in lines 181-3 needs to be revised. Are there any implications of this apparent mistake for later model calculations? I would expect the U5 and primigravid sensitivities to be about the same based on other approaches to these populations, and their relative immunity.

This was indeed a typo and we thank the reviewer for this excellent spot which eluded all co-authors through many revisions! The blue lines are those which come from the Wu et al estimate for sensitivity in young children whereas the green lines are those which come from adults. The reviewer is therefore completely correct that that sensitivity is much higher within the younger population and much closer to that in adults. Indeed, the aim of including this indication of the higher sensitivity for children was to provide context for just how high sensitivity in primigravid in their first visit during the second trimester – with the estimates of sensitivity still substantially higher than the dashed blue line, particularly in areas of lower transmission.

3. Line 199-201: I do not follow how differences in non-pregnancy specific immunity contribute to the decrease in PCR prevalence by immunity at enrolment (I know of no evidence that this is gravidity dependent). The better explanation is immunological memory related to pregnancy specific immunity, which has potentially been boosted by new exposures from 12- 20 or 24 weeks.

We agree with the reviewer that this was not particularly clearly worded but believe our model is consistent with the point the reviewer makes. The higher non-pregnancy specific immunity is because women of higher gravidity are on average of higher age and therefore likely to have increased exposure-driven immunity outside of pregnancy and these age-dependent effects on PCR prevalence are captured in the model. From towards the end of first trimester (i.e. 12 weeks) onwards our model than also captures increased immunity to limit placental sequestration (and feasibly clear below PCR thresholds by the 20-24 week

mark) along the lines the reviewer is suggesting which also contributes to reducing PCR prevalence. We have now clarified this within the text.

4. Line 238: what is the basis for the statement that DP and SP offer similar post-treatment prophylaxis? It does not appear apparent for the references, and the combination of the longer half life of piperazine and the prevalence of low grade markers of SP resistance even in "sensitive" areas (which affect MIC and this prophylaxis, White N PloS Med 2005) makes these drugs less likely to be equivalent that the authors assume.

To our knowledge there are no data that directly compare DP and SP longevity in pregnancy in areas in the absence of quintuple mutation. Instead the basis for our assumption that SP and DP offer approximately equivalent protection is that DP has been estimated by Okell et al to prevent more than 50% of infections for 29.4 days [range 16.4–48.8 days] outside of pregnancy, whereas efficacy studies with SP in pregnancy in areas where the quintuple mutation is largely absent also point to very high protection out to 28 days but with reinfections starting to occur by 42 days. That SP retains effective prophylaxis outside of pregnancy for approximately a month in areas of low quintuple mutation is also supported by estimates of SP-AQ efficacy in Seasonal Malaria Chemoprevention. Although this involves partnering SP with AQ this window of protection is substantially greater than the ~14 days of protection AQ would offer in isolation. At the same time, we are very aware of the lack of data about the relative effectiveness of these two drugs in areas of low quintuple resistance during pregnancy, which itself was a factor in our decision to not compare the two regimens directly in areas of low quintuple resistance. The data underpinning our assumptions has now been more carefully described in the methods and we have also highlighted this as a limitation in the discussion, as well as pointing to the need for more data to better inform the relative merits of DP as an alternative regimen in such settings.

5. Figure 4 – IPTp with AL might help make a point in the simulations, but it is not a real world alternative strategy for IPTp and it is debatable whether it should be included. If it is, the text needs to indicate it is simply there to illustrate how drug half life affects the models.

All of our analysis supports that a longer-lasting drug such as DP should be prioritised over AL and we have now emphasised this to a greater extent. In this analysis we use AL as a real-world example of a short-lasting drug. Moreover, although it hasn't been used for IPTp, it has been used in recent trials of ISTp (e.g. the multi-country trial in West Africa) so we believe it is worthwhile making the explicit point that it generally is a less suitable drug for routine ANC-based malaria protection.

6. Line 319 onwards: we may not have enough data to model the effects of hs-RDTs, as the authors point out the tiny studies to date do not show these tests are much better

than standard RDTs in pregnancy, so trying to develop models which take this into account may indicate they have few advantages, while models that assume they have a 10 fold lower limit of data will not fit with the “lived experience” of such tests to data.

The reviewer is entirely correct that there is insufficient data on incremental accuracy of recently developed hs-RDTs in pregnancy. Our aim in this section was not to model these tests explicitly, instead it was to consider whether a more sensitive test would ever have the potential to substantially reduce exposure in pregnancy (hence our use of a ‘perfect’ test for comparison in Figures 5 and 6. We agree this wasn’t sufficiently clarified before showing the results of this analysis so have now made this aim much more explicit when introducing it. We have also reiterated this within the relevant section of the discussion.

7. Line 399 onwards: the “intrinsic value” of IST is not really established – it is a lot more expensive than IPTp with SP and the countries where it might be deployed are presumably low transmission areas- while much of the modelling presented focuses on higher transmission areas. The data seem to illustrate why IST has not been endorsed by WHO rather than making a case for its being considered. In the case of complete failure of SP, DP IPTp would be a clearly better alternative as several figures show.

We wholeheartedly agree with the reviewer that our results support the lack of endorsement for ISTp over and above IPTp-SP. Our phrase ‘intrinsic value’ was not intended to give this impression but we can see how the word ‘value’ could be interpreted in this manner and may provide the misleading impression that we are also considering costs. Instead our aim was to highlight that the small differences in birth outcomes between IPTp-SP and ISTp are due to them both providing a level of protection that, whilst imperfect, is substantially greater than no intervention at all. As a result, we have altered our phrasing from ‘intrinsic value’ to ‘intrinsic impact relative to no intervention’. The interpretation with respect to the results of our modelling IPTp-DP was one which produced substantial discussion within the authorship team, with some sharing the view of the reviewer and others pointing out that DP has not yet been declared suitable by WHO for use in IPTp and that there are other issues concerning the use of DP as an IPTp drug (such as safety, adherence, cost and implications for first-line therapy) that were not considered in the analysis. As a result of these discussions, and given there are ongoing trials of IPTp-DP which are likely to provide additional insight, a consensus was reached that the potential increased impact of IPTp-DP in these settings should be presented and, the need for long-lasting replacement for SP highlighted, but that it was beyond the scope of this paper to draw conclusions as to whether DP is the definitive drug to serve this purpose.

Minor comments

1. Line 171 is AIC defined somewhere?

This has now been defined.

2. line 208 tretment for treatment

This has now been corrected.

3. Line 296-8 the sentence is incomplete.

This has now been corrected.

4. Figure 5- the title is not truly reflective of the message. Looking at Fig 5 a one would think that no drug would fail to clear 15% of infections at low EIR. Different wording might better reflect the message here.

This has now been clarified.

5. Line 580-1- see other comments about equivalence (or not) of prophylaxis from DP and SP.

This has been amended in line with previous comments

6. References 8, 11, 14, 15 have incomplete citations

7. References 6 and 41 are duplicates.

This has been corrected

8. Supplementary material P 6 placental infections: line 5-7. Replace with "... expressing VAR2CSA, the Plasmodium falciparum erythrocyte membrane protein 1 variant which binds to chondroitin sulphate A"

We are very grateful to the reviewer for aiding the first author with his shaky grasp of immunology!

9. Same page point 1) sequestered, not sequested

This has been corrected.

10. Page 8: I am unclear what is meant by "rate of progression through past stage infection". As far as we know this is an end stage event- it does not resolve.

A rate of progression from 'past' infection was included within the initial model fitting because it was assumed impossible to rule out that a fraction of infections which leave appreciable pigmentation immediately after clearing. Subsequently the fitted posterior distribution of the rate of progression is low enough to equate to a very large majority of 'past' infections not clearing by delivery so is largely consistent with it being an end stage

event. However, to avoid confusion, and as we already state earlier in the SI that this parameter was not relevant to this analysis, we have removed all references to a rate of progression through past stage.

11. Page 11 sentence "We therefore developed as model of RDT sensitivity at subsequent visits as a using logistic regression." needs editing

This has been amended to "We therefore modelled RDT sensitivity at subsequent visits separately using logistic regression."

12. Paragraph starting "Defining"- sentence meaning unclear.

This has now been clarified to: "We define the set of ANC visits $V = \{v \in 1..n_v\}$, with associated timings for each ANC visit t_v . The modelled sensitivity of RDT if provided at each visit, which depends upon the gravidity, visit number and previous infection status of a women at the visit as described above, is then denoted s_v ."

Reviewer #2 (Remarks to the Author):

The data analyses appear to have been carried out very competently.

A major issue that is barely addressed is the practical difficulties of implementing the different strategies in the challenging setting of MCH clinics. Diagnostic testing and/or differentiated drug regimes for different stages of pregnancy or parity represent significant complications to clinic operations. The paper does not engage with the question of the likely compliance and adherence to alternative drug regimes until line 438. Similarly, the section beginning line 344 does not address the issue that many women will not know that they are pregnant during the first trimester.

The reviewer is correct to highlight that the practicalities of changing and implement ANC based strategies are extremely important considerations. This forms a major aspect of the discussion of the work – and would like to reassure the reviewer that, as with any manuscript, the order of appearance is not the order of importance! – however we have provided more context around these issues within the introduction as we also believe this serves to further highlight the importance of modelling such as this to identify the extent to which interventions, even when delivered perfectly, can improve upon the status quo.

The capacity and resources necessary to provide any new intervention inevitably depend largely upon the context and capturing these financial and logistical constraints accurately relies upon a range of analyses and methodologies that are beyond the scope of this analysis. However none of the interventions we model are in any way hypothetical – ISTp

has been the focus of a wealth of research, whereas a hybrid strategy has already been implemented successfully nationwide in Tanzania with very high uptake (particularly given the bleak history of uptake of malaria interventions at ANC!).

The experience of ISTp is also informative in that substantial resources were invested in assessing the cost-effectiveness and provider- and user-acceptability of an intervention which was subsequently doomed by its inability to demonstrate superior per-protocol effectiveness. This helps to highlight the benefit of a more up-stream approach such as this which attempts to help identify those have potential to improve outcome as a precursor to considering large-scale trial and implementation studies which can then address issues of costs and logistics more comprehensively.

Finally, though completely anecdotal, when discussing the increased complexity of the intervention with my Tanzanian collaborators within the NMCP for our recent Lancet GH paper they contrasted the 'hybrid' approach of screening at first ANC regardless of symptoms and providing an ACT if positive with the previous policy of case management - screening on the basis of symptoms and then testing -as being, in many ways, a diagnostic algorithm with fewer steps to follow. Clearly this wouldn't be the experience for every visit or in every setting but does appear to caution against coming to one-size-fits-all conclusions about the level of incremental complexity of different ANC-based strategies.

(Figure 2b shows fitted values of an odds ratio, but it is not explicit in the legend what is the outcome being modelled).

This has now been clarified.

Reviewer #3 (Remarks to the Author):

This is an important and timely topic given a number of new trials, changing epidemiology of malaria, and dynamic patterns of SP resistance.

The case for relating more easily measured endpoints (e.g. parasite prevalence + density) to impact through modeling is compelling described in the introduction (L87-95).

This is especially useful in the handling of prospective scenarios around treatment with ACT in early pregnancy (L344 + Fig 7a/b).

We thank the reviewer for their positive feedback.

L123 + Fig. 1: An earlier statement of how enrollment worked in these trials would be helpful for context. Later, on L193, this is clarified in the introduction of the modeling work (Fig. 3)

Recruitment criteria with respect to timing of enrolment has now been added to this section.

L139-141: Again, the reader first encounters some odds ratios that would benefit from an earlier and more detailed description of how the possible differences in age distributions of enrolled primigravidae compare to the general 15+ population, and the expected scale of exposure- or immunity-related impact that could have (introduced at L178-180). The young age-group result is compelling, but one wonders whether -- in conjunction with "unexplained between-site variation" (L175) -- a more complete picture can be told.

The reviewer is correct that the use of the 15+ age range may influence our results to an extent, something we acknowledge and discussion within the same section. Unfortunately surveys rarely report sensitivity in more precise bounds such as 15-20 year-olds and sample sizes in these ages are often small which would make a more granular comparison likely to be subject to much uncertainty in the out-of-pregnancy sample. However, as the reviewer notes, our estimate that sensitivity is substantially higher in this population than in young children supports that this effect is being driven to a large extent by pregnancy, as does the recent findings we extensively cite following a nulliparous cohort of sub-RDT infected women in Benin, where densities were substantially boosted following conception.

Fig 2a: The varying gravidity axes by panel was a bit confusing, but I suppose there's an aesthetic trade-off with leaving a bunch of white space in 4 panels if they're all the same scale.

We agree there are disadvantages with both approaches to presenting this figure the reviewer describes – we decided this approach made it easier to compare the model fit to the data. Panel 2c then provides an indication of how the modelled gravidity specific patterns of detectability vary across a continuum of transmission settings.

L175 + Fig 2c: What role does the seasonality of enrollment of the gravid cohorts play in shifting the RDT/PCR relationship across sites? That might be relevant in Mali and Burkina -- both in projecting the results + in the operational opportunities available to different intersections of gestation (9 months) and high-season (3-4 months).

This is an excellent point. Recruitment was year-round in these settings so our results encompassing both seasonal peaks and nadirs in transmission in all trial-sites. The cross-sectional data in non-pregnant individuals also do not provide information on transmission season and the currently our focus has primarily been upon strategies that would be implemented uniformly throughout the year at ANC. Given the length and

complexity of the manuscript we have not incorporated the role of seasonality within this analysis, instead noting it as an important limitation. However, given the current drive for providing seasonally targeted interventions such as SMC and discussions about extending age-ranges beyond young children, we aim to look at this in depth in future work to consider whether clearing the parasite reservoir pre-conception during the dry season could have a large effect in alleviating subsequent malaria in pregnancy burden.

L392: Is there a typo in this sentence with the acronyms?

Indeed there was, we thank the reviewer for this excellent spot!

REVIEWERS' COMMENTS:

Reviewer #1 (Remarks to the Author):

I appreciate the authors detailed and thoughtful responses to my initial critique.

I have some minor additional comments or questions for the authors.

One of my only remaining comments is in relation to Figure 6 b. This claims that average numbers of new infections occurring in trimesters 2 and 3 are up to 10 in a situation with an EIR of 1, 25 with an EIR of 10 and 50 with an EIR of 100. If this means new infections per woman (and a description of what I mean by "average number of infections" is not provided in the figure legend) this seems implausible. An EIR of 1 implies one new infectious bite over a year, for example. The Figure legend should make sense as a stand alone.

In Figure 7 part a. Units for X axis would be helpful.

Figure 7 d. What is the significance of the black/grey boxes representing IPTp and no IPTp? It seems that the different colour intensity refers to precision around the estimates of LBW prevented.

References 11, 17 and 21 appear incomplete

Reviewer #2 (Remarks to the Author):

I judge that the authors have satisfactorily addressed my earlier comments

Reviewer #3 (Remarks to the Author):

The authors have adequately addressed the minor concerns raised in the initial review, and I would recommend publication.

Response to REVIEWERS' COMMENTS:

Reviewer #1 (Remarks to the Author):

I appreciate the authors detailed and thoughtful responses to my initial critique.

I have some minor additional comments or questions for the authors.

One of my only remaining comments is in relation to Figure 6 b. This claims that average numbers of new infections occurring in trimesters 2 and 3 are up to 10 in a situation with an EIR of 1, 25 with an EIR of 10 and 50 with an EIR of 100. If this means new infections per woman (and a description of what is meant by "average number of infections" is not provided in the figure legend) this seems implausible. An EIR of 1 implies one new infectious bite over a year, for example. The Figure legend should make sense as a stand alone.

This was indeed an axis labelling error moving from the % in Figure 6a to the average number of infections in Figure 6b – this has now been corrected and should now hopefully make a lot more sense! We thank the reviewer for their excellent spot!

In Figure 7 part a. Units for X axis would be helpful.

It has now been clarified this refers to days

Figure 7 d. What is the significance of the black/grey boxes representing IPTp and no IPTp? It seems that the different colour intensity refers to precision around the estimates of LBW prevented.

We have now clarified in the figure legend that this is to illustrate our uncertainty in the extent to which 'catch-up' growth can occur if women are better protected from exposure later in pregnancy.

References 11, 17 and 21 appear incomplete

These have now been corrected

Reviewer #2 (Remarks to the Author):

I judge that the authors have satisfactorily addressed my earlier comments

Reviewer #3 (Remarks to the Author):

The authors have adequately addressed the minor concerns raised in the initial review, and I would recommend publication.

We thank all three reviewers for their incredibly helpful and constructive reviews.